# Impact of Impaired Kidney Function on Arrhythmia-Promoting Cardiac Ion Channel Regulation

**DOI:** 10.3390/ijms241814198

**Published:** 2023-09-17

**Authors:** Frederick Sinha, Frank Schweda, Lars S. Maier, Stefan Wagner

**Affiliations:** 1Department for Internal Medicine II, University Medical Center Regensburg, 93053 Regensburg, Germany; frederick.sinha@ukr.de (F.S.);; 2Institute of Physiology, University of Regensburg, 93053 Regensburg, Germany

**Keywords:** arrhythmia, chronic kidney disease, cardio renal, SGLT2 inhibitor, CaMKII, cardiac electrophysiology, renin–angiotensin–aldosterone, acidosis, autonomic nervous system, connexin 43

## Abstract

Chronic kidney disease (CKD) is associated with a significantly increased risk of cardiovascular events and sudden cardiac death. Although arrhythmias are one of the most common causes of sudden cardiac death in CKD patients, the molecular mechanisms involved in the development of arrhythmias are still poorly understood. In this narrative review, therefore, we summarize the current knowledge on the regulation of cardiac ion channels that contribute to arrhythmia in CKD. We do this by first explaining the excitation–contraction coupling, outlining current translational research approaches, then explaining the main characteristics in CKD patients, such as abnormalities in electrolytes and pH, activation of the autonomic nervous system, and the renin–angiotensin–aldosterone system, as well as current evidence for proarrhythmic properties of uremic toxins. Finally, we discuss the substance class of sodium–glucose co-transporter 2 inhibitors (SGLT2i) on their potential to modify cardiac channel regulation in CKD and, therefore, as a treatment option for arrhythmias.

## 1. Introduction

Chronic kidney disease (CKD) is defined as a glomerular filtration rate (GFR) < 60 mL/min/1.73 m^2^ or kidney damage, which can be represented by an albumin-to-creatnine ratio >30 mg/g for more than 3 months [1]. Impaired kidney function increases the prevalence of cardiovascular complications and mortality, especially for arrhythmias. The prevalence and severity of arrhythmias increase with declining kidney function. In patients with end-stage renal disease (ESRD), sudden cardiac death (SCD) accounts for between 40 and 60% of all deaths [2,3,4]. However, the cellular mechanisms of arrhythmia in chronic kidney disease are not fully understood. In this review, we briefly summarize the current knowledge of cardiac ion channel regulation contributing to arrhythmia in chronic kidney disease. First, we briefly describe the most important channels in cardiomyocytes that contribute to excitation-contraction coupling. Then, as summarized in Figure 1, we discuss the impact of changes in electrolytes, pH homeostasis, autonomous nerve system, and renin–angiotensin–aldosterone regulation on ion channel function in CKD patients. Finally, we discuss SGLT2 inhibitors as an evolving strategy for the treatment of arrhythmias in CKD patients.

### Excitation-Contraction Coupling in Healthy Cardiomyocytes

Cardiac excitation–contraction coupling (ECC) is based on the activation and inactivation of different ion channels, which causes a calcium-dependent contraction. The excitation is initiated by Na^+^ influx via voltage-gated fast Na^+^ channels (Na_V_1.5), which depolarize the membrane up to +20 mV (phase 0). The positive membrane potential induces the fast inactivation of the Na_V_1.5 channels and the activation of transient outward K^+^ channels (K_V_1.4, K_V_4.2, K_V_4.3). This transient outward K current (I_to_) results in an early repolarization (phase 1), which is followed by a plateau phase (phase 2). The plateau phase describes a balance between the small K^+^ current flux and the Ca^2+^ inward current (I_Ca_) by slow-activated, voltage-gated L-type Ca^2+^ channels (Ca_V_1.2) working in tandem with the late sodium current (Late I_Na_). This balance can be easily disturbed even by small currents that destabilize the plateau membrane potential, causing arrhythmias. In phase 3, Ca^2+^ channels progressively reduce their conductance, and slow K^+^ channels (K_V_7.1 and Kir2.x channels) activate, which results in the repolarization of the membrane potential [5].

During the cardiac action potential, Ca^2+^ entry triggers ryanodine receptor (RyR2) opening and, therefore, Ca^2+^ release from the sarcoplasmic reticulum (SR). Increasing the cytoplasmic Ca^2+^ level allows Ca^2+^ to bind to the myofilament protein troponin C, which then results in contraction during systole. During diastole, SR Ca^2+^-ATPase (SERCA2a) and Na^+^/Ca^2+^ exchange (NCX) remove the vast majority of the cytosolic Ca^2+^, and a tiny fraction of Ca^2+^ can be removed from the cytosol by the sarcolemmal Ca^2+^-ATPase and mitochondrial Ca^2+^-uniporter [5].

Ventricular arrhythmias are the result of electric instability in cardiomyocytes caused by cell membrane hyperexcitability, altered repolarization, or disturbed conduction of the electrical wavefront across the myocardium [6]. There are two basic mechanisms by which arrhythmias occur in the heart: re-entry arrhythmia and triggered activity. Re-entry occurs when an excitation spreads around an obstacle and hits tissue that has been previously excited. The obstacle can be a structural change, such as an infarct scar or a functionally altered tissue. Functional limitations of the conduction velocity or conduction block can result, for example, from a decrease in the conductivity of the cell–cell contact (e.g., Connexin 43) at the intercalated discs or prolonged recovery time, facilitating life-threatening re-entry arrhythmias [7,8].

Triggered activity can be caused by an imbalance of ion currents in a single cardiomyocyte. Depending on the mechanism of occurrence, these imbalances can be classified as early after-depolarizations (EADs) or delayed after-depolarizations (DADs).

EADs occur during the action potential and are often the result of disproportional fast depolarization or attenuated repolarization. In particular, the plateau phase is vulnerable to EADs, as even small imbalances between Ca^2+^ influx and K^+^ efflux can cause a new depolarization with SR calcium release. This can lead to life-threatening tachyarrhythmia if it occurs in several cardiomyocytes simultaneously.

DADs are a product of cytosolic or SR calcium overload. SR calcium overload leads to the spontaneous release of calcium sparks in the cytosol, which then activate the sodium–calcium exchanger (NCX) to remove Ca^2+^ from the cytosol. The NCX imports three Na^+^ for each Ca^2+^ excreted, causing a net import of positive charge, and it can thus depolarize the membrane potential. This can again trigger an AP, which leads to a renewed activation of the calcium influx into the cardiomyocytes, which under pathological conditions can trigger a ventricular arrhythmia [6].

## 2. Translational Approaches to Arrhythmia in CKD

As kidney function declines, the risk of atrial and ventricular arrhythmias increases. A cross-sectional analysis from the Atherosclerosis Risk in Community study (ARIC) showed in >2000 patients with GFR < 60 mL/min/m^2^ within 2 weeks of cardiac monitoring a high prevalence of ectopic beats in >90% of the patients, ventricular tachycardia (30.2%), and atrial fibrillation (7.4%) [9]. In patients with end-stage renal disease (ESRD) on dialysis, the prevalence of ventricular arrhythmia increased, as 25.4% of patients experienced asystole or bradyarrhythmia, and 23% of the patients had tachyarrhythmia [10]. The bulk of clinical data does not allow differentiation of the specific signaling pathways of cardiac ion channel regulation.

Therefore, in vivo models in rodents and in vitro models with isolated cardiomyocytes have been established. The most common in vivo model for CKD being used to investigate arrhythmia is subtotal nephrectomy (SNx) [11]. The abrupt reduction in the number of glomeruli is accompanied by an accumulation of urinary substances, electrolyte and pH imbalances, and compensatory mechanisms, such as the activation of the renin–angiotensin–aldosterone system (RAAS) and autonomic nervous system (ANS). Further models include adenine feeding, resulting in tubulointerstitial fibrosis, and genetic modification, such as Col4a3 deficiency, leading to progressive glomerulonephritis with microhematuria and proteinuria [12]. Another approach is the incubation of isolated cardiomyocytes with uremic plasma [13,14,15]. Although these approaches showed seminal changes in action potential and dysbalanced calcium handling, explaining the arrhythmogenicity in chronic kidney failure, more sophisticated approaches are needed to distinguish which mechanism or metabolite is involved in ion channel regulation [16] (Figure 2). Therefore, in the following sections, we present the regulation of cardiac ion channels based on four specific changes in patients and models of CKD.

## 3. Characteristics of Chronic Kidney Disease

### 3.1. Electrolyte and pH Abnormalities in CKD

The kidneys play an essential role in the regulation of serum potassium levels and pH. In CKD, the capacity of potassium and H^+^ secretion declines with reduced glomerular filtration rate, resulting in electrolyte and pH imbalance [17]. The most common abnormalities in CKD are hyperkalemia and metabolic acidosis [17].

During hyperkalemia, the physiological gradient of high intracellular and low extracellular potassium is attenuated. According to the Nernst equation, the resting membrane potential depolarizes due to a reduction in the K^+^ gradient. Depolarization of the membrane potential initially leads to increased excitability due to the membrane potential being closer to the activation potential of cardiac voltage-gated Na_V_1.5 channels. However, with further depolarization of the membrane potential, the steady-state inactivation of Na_V_1.5 channels also increases, leading to fewer channels contributing to AP upstroke during every depolarization [18]. This loss of peak Na^+^ channel function reduces the amplitude of the Na^+^ peak and conduction velocity in the myocardium, which is electrocardiographically represented by a broad QRS complex in hyperkalemia [19]. Reduced conduction velocity increases the risk that the action potential wave front will excite already recovered tissue, again leading to re-entry arrhythmias. Additionally, the action potential duration of the myocardium decreases in hyperkalemia. This effect can be explained by an increased repolarization reserve, as hyperkalemia facilitates the inward K^+^ current (I_K1_), one of the main repolarization currents in phase 3 of the AP. This can be clinically observed as alternans in T-waves on ECG in hyperkalemia [20]. Enhanced I_K1_ can also induce post-repolarization refractoriness, which prolongs the period of inexcitability of cardiomyocytes [21]. Spatially discordant excitability of cardiomyocytes and APD alternans in amplitude and duration result in conduction velocity alternans that predispose to re-entry arrhythmias [18,22].

The effect on cardiac ion channels in metabolic acidosis is complex. Protons directly inhibit the sodium conductance of the Na_V_1.5 in cardiomyocytes by changing the electrostatic potential of the channel with the protonation of carboxyl residues. This results in a reduced peak sodium current with reduced conduction velocity [23]. In CKD patients with acidosis, this effect was clinically manifested as the prolongation of the QT interval on ECG, which was corrected by the treatment of the acidosis with bicarbonate administration. Additionally, acidosis was found to destabilize the fast-inactivated state of Na_V_1.5 and increase persistent I_Na_ in isolated canine cardiomyocytes [24,25]. The decreased peak and persistent I_Na_ lead to prolonged repolarization, which can trigger calcium release as early after-depolarizations (EADs) [6].

Acidosis is known for reducing inotropy of cardiomyocytes [26,27]. This effect can be explained by molecular experiments in isolated rat ventricular cardiomyocytes that showed desensitized RyRs, resulting in decreased SR Ca^2+^ release combined with desensitized myofilaments. Additionally, experiments on isolated rat ventricular cardiomyocytes showed that systemic acidosis also reduced cytosolic pH. In these cells, acidosis stimulated acid extrusion via Na^+^/H^+^ exchange (NHE), which subsequently increased intracellular Na^+^ and thereby lowered Ca^2+^ efflux via NCX [28]. The combination of decreased SR Ca^2+^ release and low Ca^2+^ efflux results in SR Ca^2+^ accumulation, which favors spontaneous Ca^2+^ release (Ca^2+^ sparks) during diastole. Although the conductance of NCX is generally reduced by acidosis [29], due to the spatial proximity of NCX and RyRs, the RyRs Ca^2+^ spark temporarily activates the NCX. Since the NCX exchanges Ca^2+^ for Na^+^ in a 1:3 ratio, the activation results in an import of a positive charge with an increase in cytosolic Na^+^ [30], which can trigger a DAD and thereby initiate an arrhythmic AP [31]. In addition, acidosis can worsen myocardial conductance by inhibiting the conductance of gap junctions, such as connexin 43 [32], which promotes the development of malignant re-entry arrhythmias [33,34].

### 3.2. Sympathetic Nerve System Activation in CKD

The sympathetic nervous system is an important regulator of blood pressure and heart rhythm, and its force is crucial for appropriate kidney function [35]. Sympathetic innervation of the heart is mainly mediated by norepinephrine and circulating adrenaline targeting β-adrenergic receptors (β-AR). On a molecular level, the stimulation of adrenergic receptors enhances cyclic adenosine monophosphate (cAMP), which enhances the conductance of cardiac ion channels either directly or by phosphorylation by protein kinase A (PKA) [36]. The sympathetic nervous activity is counteracted by the parasympathetic nervous system, which is mainly mediated through acetylcholine binding to muscarinic receptors. While β-AR receptors are present throughout the entire myocardium, muscarinic receptors in the heart are essentially present in the atrium, sinoatrial, and AV node. The activation of muscarinic receptors results in nitric oxide (NO), which stimulates cGMP levels. Increased cGMP levels finally enhance phosphodiesterases that break down cAMP and thereby oppose the sympathetic stimulation of the atrium and conduction system of the heart [37].

In CKD, clinical data and rodent experiments have shown a sympathetic overdrive assessed by muscle sympathetic nerve activity and serum levels of norepinephrine [35,38,39]. Although the mechanisms are not fully elucidated yet, patients with CKD showed a significant increase in inflammation, with oxidative stress leading to a reduced bioavailability of NO [40]. Preclinical and clinical investigations identified a reduced level of endothelial NO as a potent central stimulator of the sympathetic nervous system [35,41]. Additionally, Converse et al. investigated the effect of bilateral nephrectomy in patients and found reduced sympathetic tone compared with patients with ESRD and healthy patients [39]. These findings suggest the direct stimulation of the sympathetic nervous system by the kidneys, which is supported by experiments with renal denervation, which decreased circulating epinephrine levels in a hypertensive pig model [42]. In CKD animal models of subtotal nephrectomy and adenine-induced CKD, renal denervation reduced atrial and ventricular arrhythmogenicity [43,44,45]. Zoccali et al. found that in a cohort of ESRD patients, the plasma level of epinephrine predicted the incidence of cardiovascular events, with arrhythmia being the second-most frequent event recorded [46]. These experiments highlight a direct link between CKD-induced sympathetic overdrive increasing the risk of arrhythmias, which raises the question of the influence of the sympathetic nervous system on the regulation of cardiac ion channels.

The activation of β-AR in cardiomyocytes increases via cAMP-dependent protein kinase A (PKA) phosphorylation of the L-type Ca channel Ca_V_1.2 (LTCC), sarcoplasmic/endoplasmic reticulum Ca^2+^ ATPase 2 (SERCA2), and phospholamban [47]. This phosphorylation results in increased sarcoplasmic Ca^2+^ levels, which enhances the systolic Ca^2+^ transient and positive inotropy. Enhanced I_Ca_ would lead to a prolongation of the APD since the plateau phase of the AP is determined by the balance between Ca^2+^ influx and K^+^ efflux. First, the PKA-dependent phosphorylation of SERCA2 and PLN counterbalances the increased influx of calcium by increasing the SR Ca^2+^ reuptake [48]. Furthermore, Bennett et al. found that in guinea pig ventricular myocytes, β-adrenergic stimulation increased the delayed rectifying potassium current (I_K_), which counteracted the AP prolongation by I_Ca_ [49]. This resulted in an accelerated repolarization and thereby shortened the AP [50].

β-AR also triggers spontaneous diastolic Ca^2+^ release and thus increases cytoplasmic Ca^2+^ levels, which activates the Na^+^-Ca^2+^ exchanger (NCX) [51]. Increased NCX activity leads to Na^+^ accumulation in cardiomyocytes, which then depolarizes the membrane to the action potential threshold. The more positive the diastolic membrane potential, the more delayed after-depolarizations (DADs) occur, which makes the cardiomyocytes susceptible to atrial and ventricular arrhythmia [52,53,54]. Besides triggered activity, beta-stimulation in rats showed that connexin 43 was redistributed [55], which is known to promote electrical instability and life-threatening arrhythmias [56].

Impaired kidney function was found to reduce parasympathetic stimulation by impaired cardiac vagal stimulation via acetylcholine (ACh) on muscarinic receptors in hemodialysis patients [57]. Muscarinic stimulation leads to hyperpolarization and shortens the APD of the atrium. This happens mainly by reducing cAMP [37] in the atrium and sinoatrial node and increasing the K^+^ conductance of the G-protein-gated inwardly rectifying potassium (GIRK1 and GIRK4) channels, as measured in isolated guinea pig atrium myocytes with the patch clamp technique [58]. Impaired parasympathetic stimulation could thus explain the tendency toward atrial rhythm disturbances, such as atrial fibrillation, in CKD patients, but ventricular arrhythmias are also present in CKD [9]. Interestingly, Liang et al. localized GIRK1 and GIRK4 in mice and human ventricles with immunohistochemistry and demonstrated that the stimulation of ventricular myocytes with ACh significantly increased the action potential duration in wild-type mice, whereas these results were not observed in GIRK4 knockout mice [59]. Furthermore, the same authors showed that in a family with LongQT syndrome, which causes life-threatening ventricular arrhythmias, the gene coding for GIRK, KCNJ5, was mutated [60]. The authors concluded that GIRKs play an important role in the repolarization of ventricular cells, which may contribute, at least in part, to the development of arrhythmias in CKD patients [59].

### 3.3. The Renin–Angiotensin–Aldosterone System in CKD

Due to the reduced renal blood flow in CKD and sympathetic nerve stimulation, renin synthesis and secretion are stimulated and thereby activate the renin–angiotensin–aldosterone cascade [61,62]. Since the activation of the RAAS plays a central role in the progression of CKD and congestive heart failure, RAAS blockade is an essential drug target in their treatment.

Angiotensin II is known to regulate several ion channels in cardiomyocytes resulting in changes in the cardiac action potential. In isolated mice ventricular cardiomyocytes, angiotensin II amplified reactive oxygen species accumulation via NADPH oxidase 2 (NOX2), which activates Ca^2+^/Calmodulin-dependent protein kinase II (CaMKII). CaMKII is known as a central regulator in failing hearts that increases the risk of arrhythmia [63,64]. CaMKII activation resulted in increased peak I_Na_ and late I_Na_ current, which can cause APD prolongation and EADs [65]. Experimental inhibition of CaMKII in isolated rabbit cardiomyocytes incubated with ROS-generated H_2_O_2_ showed a reduction in EADs [66].

Also, in rat ventricular myocytes, angiotensin II and aldosterone both inhibited the Ca^2+^-independent transient outward K^+^ current (I_to_), which resulted in action potential prolongation, which is known to be proarrhythmogenic [67,68].

While aldosterone increases L-type Ca^2+^ current in rat ventricular cells by upregulating transcription, the activity of Ca_V_1.2 did not change with acute and chronic incubation of cardiomyocytes with aldosterone [69]. However, angiotensin II additionally seems to induce an increased I_Ca_ by increasing the activity of Ca_V_1.2 [70,71,72]. One possible mechanism of the increased activity of the I_Ca_ current is the stimulation of the CaMKII. CaMKII facilitates the phosphorylation of LTCC, phospholamban (PLN) at Thr17, and ryanodine (RYR) at site S2814 [73,74]. These hyperphosphorylations increase spontaneous diastolic Ca^2+^ release, which again can trigger atrial and ventricular arrhythmias [65]. Therefore, the inhibition of CaMKII is a possible target to reduce arrhythmias in heart disease [75].

### 3.4. Uremic Toxins

With the progression of CKD and GFR decline, compounds accumulate in patients’ blood; under healthy conditions, they are filtered and excreted by the kidneys, the so-called uremic toxins. The large number of uremic toxins can be classified into several chemical groups by their molecular weight, origin, and molecular targets, and some of them are well-established as biomarkers in early-stage chronic kidney disease [76].

In vivo and in vitro experiments have aimed to characterize the impact of uremic conditions on cardiac electrophysiology and identify uremic toxins that have direct alterations of cardiac ion channels.

Indoxylsulfate (IS) is a metabolite of L-tryptophan that accumulates with a declining filtration rate [77]. In patients with CKD, IS has been independently associated with a prolonged QT interval, which increases the risk of arrhythmias [78,79]. On a molecular level, van Ham et al. conducted in vitro experiments with the incubation (48 h) of human induced pluripotent stem cell-derived cardiomyocytes (hiPSC-CMs) and immortalized HEK293 cells with IS. They found decreased repolarizing potassium currents (I_Kr_) and lowered protein levels of K_V_11.1, which subsequently resulted in prolonged action potential duration [80,81]. Furthermore, indoxyl sulfate decreased gap junction intercellular communication by downregulating connexin 43 in cultured neonatal rat cardiomyocytes, leading to increased spontaneous contractions [82]. Another uremic toxin, p-cresyl sulfate, also decreased the potassium channel K_V_1.2, which, in computational modeling, prolonged the action potential [81]. Chen et al. showed that in pulmonary veins isolated from rabbits, IS induced increased calcium leakage, resulting in an increased occurrence of delayed after-depolarizations [83].

Fibroblast growth factor 23 (FGF23) is a hormone for the homeostasis of phosphate and calcitriol synthesized in osteoblasts and osteocytes. In the early stages of CKD, FGF23 increases, with independent associations with atrial arrhythmias and ventricular hypertrophy both increasing mortality in clinical studies [84,85]. FGF-23 alters Ca^2+^ handling in cardiomyocytes via increased RyR2 and PLB phosphorylation at the PKA and CaMKII sites, which increases SR Ca^2+^ spark frequency. According to Navarro-Garcia et al., this Ca^2+^ mishandling induces arrhythmic events, as assessed in mice with electrocardiographic monitoring, with an increased rate of premature ventricular contractions (PVC) after intraperitoneal FGF-23 injection. Additionally, ex vivo FGF-23 perfusion of mice hearts showed a prolonged QT interval [86], and in vitro incubation of ventricular myocytes from mice with FGF-23 increased the number of EADs and DADs [12,87]. The signaling pathway of FGF-23 is mediated by the reduction of AMP-specific 3′,5′-cyclic phosphodiesterase 4B (PDE4B), which leads to cAMP-dependent phosphorylation of calcium RyR2 and PLB. Klotho, a proteohormone counteracting FGF-23, diminished calcium imbalance by reducing diastolic SR Ca^2+^ release and thereby proarrhythmic events [88].

### 3.5. SGLT2- Inhibitors: Novel Treatment Strategies for Arrhythmia in CKD?

SGLT2 inhibitors have fundamentally expanded heart failure and CKD therapy in recent years. Due to the previously known mechanism of action, the blockade of SGLT2 in the proximal tubule of the kidney with glycosuria, SGLT2 inhibitors were initially used for diabetes therapy. However, large studies in patients with heart failure independent of ejection fraction and in patients with both diabetic and non-diabetic chronic kidney disease showed a reduction in mortality and cardiovascular complications [89,90,91,92,93,94].

A meta-analysis of 34 clinical trials with more than 100,000 patients included showed a decrease in atrial and ventricular tachyarrhythmias by SGLT2 inhibitors [95,96,97,98]. However, because of the heterogenicity of the experimental groups, no statistical significance was found for a reduced risk of arrhythmias in CKD without HF [95]. Nonetheless, SGLT2 inhibitors were shown in clinical studies and experimental work to act both via the improvement of the characteristics of CKD listed above and directly on the myocardium.

In a meta-analysis of a total of 49,000 patients with type II diabetes, the SGLT2 inhibitor Empagliflozin significantly reduced the risk of severe hyperkalemia (≥6.0 mmol/L, hazard ratio (HR) 0.84, *p* < 0.001) in patients with an eGFR < 60 mL/min/m^2^ without increasing the risk of hypokalemia [99]. This mechanism can be explained by the increased natriuretic and diuretic effects of SGLT2 inhibitors [99,100]. Evidence from neurogenic hypertensive mice fed a high-fat diet showed that treatment with the SGLT2 inhibitor Dapagliflozin decreased norepinephrine levels in the same manner as pharmacological sympathetic nerve inhibition did [101]. Although there is no evidence in CKD that Empagliflozin directly inhibits RAAS [102,103], there are clinical data underlining the beneficial effect of the combination of both ACEi and SGLT2i, suggesting a synergistic effect [104].

Besides the indirect effects, there is evidence that SGLT2 inhibitors directly regulate cardiac ion channels (Figure 3). SGLT2 inhibitors are potent correctors of pathological action potential changes. In mouse models, in which the QT interval was prolonged by either myocardial infarction or amitriptyline administration, Empagliflozin reduced the QT interval and significantly reduced EADs [105]. In another mouse model with a prolonged QT time by the induction of metabolic syndrome, Jhuo et al. found that connexin 43 expression was reduced in the ventricle. SGLT2i treatment increased the expression of connexin 43, which was accompanied by a significant shortening of the QT interval, highlighting SGLT2i as a treatment option for malignant re-entry arrhythmias in CKD [7,106]. These findings are in line with data from Dos Santos et al., who demonstrated APD duration normalization by SGLT2i after hypoxia-induced APD prolongation in iPSC [107]. Increased late sodium current (I_Na_) is a well-known mechanism of prolonged APD, with an increased risk of EADs [108] leading to ventricular arrhythmia. Indeed, Mustroph et al. showed that in mice and patients with reduced ejection fraction causing an increased late Na^+^ current, Empagliflozin inhibited the late Na^+^ current by inhibition of the CaMKII [109,110]. Additionally, Empagliflozin reduced spontaneous diastolic SR Ca^2+^ release by the inhibition of RyR2, which was accompanied by a reduction in CaMKII activity [111,112]. All in all, SGLT2i demonstrated antiarrhythmic properties by improving the above-mentioned four mechanisms, which occur in CKD. Additionally, SGLT2i inhibitors seem to inhibit phosphorylation by CaMKII, although the direct mechanisms have not been elucidated yet.

## 4. Conclusions

Chronic kidney disease is a common and complex disease in which the most frequent causes of death are cardiovascular events such as arrhythmias. In particular, changes in electrolytes, pH, the activation of the autonomic nervous system (ANS), the renin–angiotensin–aldosterone system (RAAS), and proarrhythmogenicity of uremic toxins contribute to arrhythmogenesis in chronic kidney disease. These adaptations to CKD, in addition to predisposing to re-entry tachycardia by conduction velocity alterations, for instance, through altered connexin 43 expression, lead to EADs via action potential prolongation and to DADs via spontaneous calcium release via the activation of the NCX. EADs and DADs increase the risk of arrhythmias. A central role in the regulation of the action potential forming and calcium handling is the activation of Ca^2+^/Calmodulin-dependent protein kinase II (CaMKII) by reactive oxygen species. This kinase is also the approach for SGLT2 inhibitors, which have already shown potential for reducing atrial and ventricular arrhythmias in experimental work. Further studies are necessary to clearly confirm SGLT2 inhibitors as antiarrhythmic agents.

## Figures and Tables

**Figure 1 ijms-24-14198-f001:**
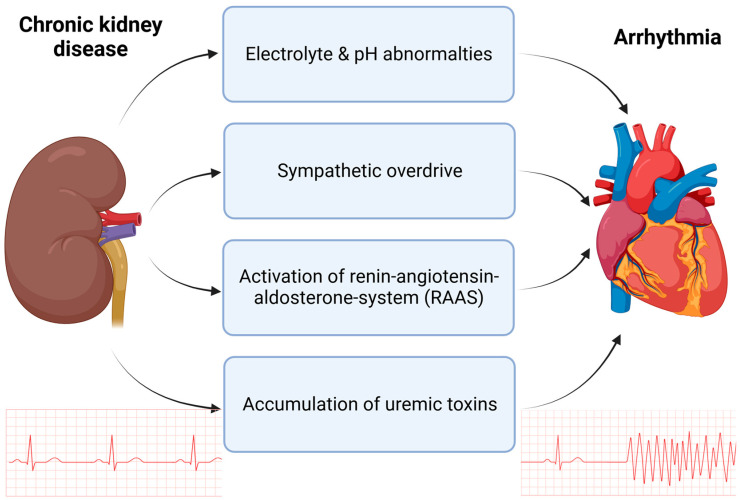
Mechanisms in CKD contributing to increased arrhythmogenesis. Chronic kidney disease is accompanied by electrolyte and pH abnormalities, such as hyperkalemia and metabolic acidosis. Additionally, in patients with CKD, an overdrive of the sympathetic nervous system is overserved as well as chronic upregulation of the renin–angiotensin–aldosterone system (RAAS). The accumulation of uremic toxins, such as indoxylsulfate (IS) and fibroblast growth factor (FGF-23), contribute to arrhythmia in CKD. Created with BioRender.com, accessed on 31 August 2023.

**Figure 2 ijms-24-14198-f002:**
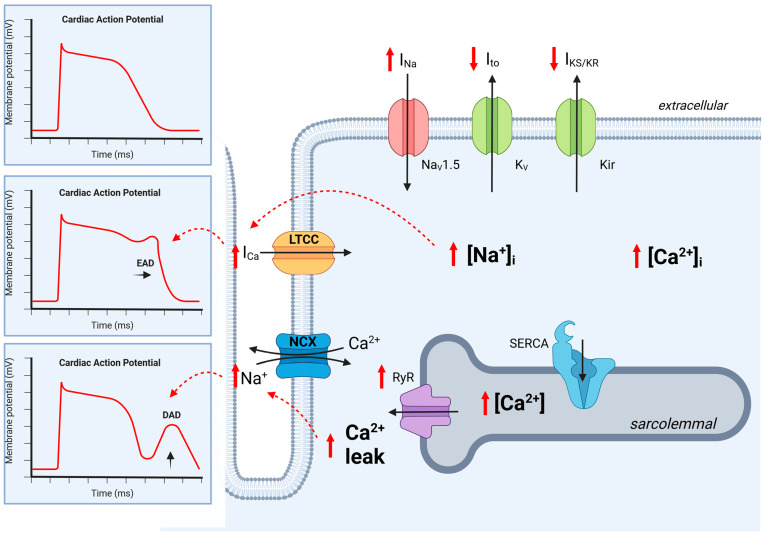
Ion channel regulation in cardiomyocytes in chronic kidney disease. The regulation of ion channels leads to abnormalities in cardiac action potential. In CKD, Na^+^ influx is disturbed and prolonged as late I_Na_, resulting in increased cytoplasmic Na^+^ levels [Na^+^]_I_ and increased Ca^2+^ influx (I_Ca_) via the L-type calcium channel (LTCC). This results in a prolonged action potential with the reactivation of the LTCC, causing an early after-depolarization (EAD). Additionally, an increased sarcolemmal Ca^2+^ concentration increases Ca^2+^ leak, which then activates NCX, resulting in delayed after-depolarization (DAD). CKD leads to the downregulation of K^+^ efflux, which prolongs the action potential duration. Dashed red lines indicate activation. K_ir_, K^+^ inward rectifier; K_V_, K^+^ efflux; Na_V_1.5, Na^+^ influx; LTCC, L-type calcium channel; NCX, Na^+^-Ca^2+^ exchanger; RyR, ryanodine receptor; SERCA, sarcoplasmic reticulum, EAD, early after-depolarization; DAD, delayed after-depolarization, black arrow, Ion current; red arrow, regulation by CKD. Created with BioRender.com, accessed on 31 August 2023.

**Figure 3 ijms-24-14198-f003:**
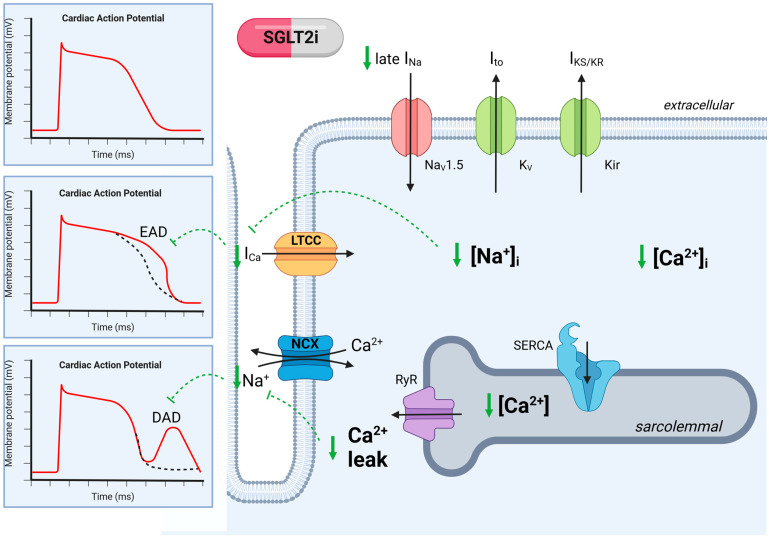
Ion channel regulation in cardiomyocytes in chronic kidney disease with SGLT2 inhibition. Regulation of ion channels leads to abnormalities in cardiac action potential, which is reversed by SGLT2i. SGLT2i reduces Na^+^ influx and late Na^+^ (I_Na_), which decreases Na^+^ overload and thereby reduces Ca^2+^ influx, which decreases the risk of action potential prolongation. SGLT2i decreases the sarcolemmal Ca^2+^ concentration, which reduces spontaneous Ca^2+^ leakage. The reduction in Ca^2+^ leak decreases the risk of delayed after-polarization (DAD). Dashed green lines indicate SGLT2i inhibition. K_ir_, K^+^ inward rectifier; K_V_, K^+^ efflux; Na_V_1.5, Na^+^ influx; LTCC, L-type calcium channel; NCX, Na^+^-Ca^2+^ exchanger; RyR, Ryanodin receptor; SERCA, sarcoplasmic reticulum; EAD, early after-depolarization; DAD, delayed after-depolarization. Created with BioRender.com, accessed on 31 August 2023.

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
