# Peer review of "Impact of Impaired Kidney Function on Arrhythmia-Promoting Cardiac Ion Channel Regulation"

_ijms, 2023, doi:10.3390/ijms241814198_

Round 1

Reviewer 1 Report

This is a very nicely written and comprehensive review about the mechanisms responsible for cardiac arrhythmias in CKD. In fact, it is really rare to come across a review that contains enough information to cover the topic but not too much information that it becomes too verbose. Really great job done by the authors. Really I only have one error I caught, please see below. Otherwise, really nicely done.

line 76: I think the authors meant delayed afterdepolarizations rather than delayed afterpolarizations?

Author Response

Reviewer #1

Authors: In detail, we have addressed the concerns voiced by the reviewers as follows:

  • This is a very nicely written and comprehensive review about the mechanisms responsible for cardiac arrhythmias in CKD. In fact, it is really rare to come across a review that contains enough information to cover the topic but not too much information that it becomes too verbose. Really great job done by the authors. Really I only have one error I caught, please see below. Otherwise, really nicely done. Line 76: I think the authors meant delayed afterdepolarizations rather than delayed afterpolarizations?

Authors: We thank reviewer 1 for his professional assessment of our article and his comment. Of course, we have corrected the error in line 76 (now line 79-80: “Depending on the mechanism of occurrence, these imbalances can be classified as early afterdepolarizations (EADs) and delayed afterdepolarizations (DADs)”).

Reviewer 2 Report

The review manuscript deals with clinically highly relevant issue, e.g., chronic kidney diseases (reviewed recently by Zoccali et al.2023) while focusing on cardiac arrhythmias risk that is challenging for further investigations. Review is perfectly written and organized, nevertheless, information about implication of myocardial connexin43 channels in arrhythmogenesis in general and in kidney diseases is missing. This topic is should be addressed in future research as well, namely the effect of uremic acid and its toxins on myocardial connexin43 expression and topology. It should be taken into consideration that ion channels disturbances trigger transient arrhythmias, while disorders of connexin channels underlie arrhythmogenic substrate for malignant re-entry arrhythmias (see f.e. Valderrábano 2007, Hoagland 2019, Saffitz 2000, Danik 2004, Salameh 2005, Dhein 2014, Thomas 2018, Dhein 2021), Besides, connexin43 hemichannels activation along with pro-inflammatory signalling may be proarrhythmic (Andelova 2021). In this context to examine efficacy of treatment with SGLT2 inhibitors would be very interesting as also noted in Zoccali et al 2023 and Andelova et al. 2022).  

Title Consider, please, to modify by including arrhythmia risk. 

Abstract exhibits several typing errors using ;  that should be deleted.

Key words: should be more precisely selected and should include connexin43

In subheading 1.1,

The text in all paragraphs should be cited with most appropriate reference. Moreover, electrical instability is promoted not only by ion channels disorders but also by disorders in myocardial conduction. The latter is strongly affected by intercellular connexin channels at the gap junction located at the intercalated disc of the cardiomyocytes (see f.e. Valderrábano 2007, Kotadia 2020)

Subheading 3.1,

Downregulation and/or inhibition of Cx43 channels contribute significantly to electrical uncoupling of the cardiomyocytes promoting slowing and-or block of conduction facilitating life-threatening re-entry arrhythmias (see f.e. Danik 2004 as well as others).

Furthermore, acidosis and Ca2+ overload also inhibit myocardial connexin43channels and contribute to electrical uncoupling of cardiomyocytes resulting in slowing of conduction that facilitates re-entrant arrhythmias (see f.e. Salameh 2005.

Subheading 3.2.

First paragraph should be more rigorously cited. Moreover, inflammation and oxidative stress downregulate conexin43 protein (see f.e. Andelova 2022 as well as others) and deteriorate the function of connexin43 channels that contribute to electrical instability and increases susceptibility of the heart to malignant arrhythmias.

Moreover, beta-adrenergic stimulation down-regulate connexin43 and alters its cardiomyocyte topology (shown by Szeiffova Bacova 2020 a,b) that is pro-arrhythmic and increase susceptibility of the heart to malignant arrhythmias (Szeiffova Bacova 2020a,b).

Conclusion should also involve possible implication of connexin43 channels in development of life-threatening re-entrant arrhythmias in CKD.

Author Response

Reviewer #2

Authors: In detail, we have addressed the concerns voiced by the reviewers as follows:

  • “The review manuscript deals with clinically highly relevant issue, e.g., chronic kidney diseases (reviewed recently by Zoccali et al. 2023) while focusing on cardiac arrhythmias risk that is challenging for further investigations. Review is perfectly written and organized, nevertheless, information about implication of myocardial connexin43 channels in arrhythmogenesis in general and in kidney diseases is missing. This topic is should be addressed in future research as well, namely the effect of uremic acid and its toxins on myocardial connexin43 expression and topology.

Authors: We are grateful to reviewer #2 for the extensive and detailed feedback on our manuscript, as we believe that this helped us improve our work. Since connexins may indeed play an essential role in the development of arrhythmias, we have included them in the review (line 74-77: “Functional limitations of the conduction velocity or conduction block can result, for example, from a decrease in the conductivity of the cell-cell contacts (e.g. Connexin 43) at the intercalated disks or from prolonged recovery time facilitating life-threatening re-entry arrhythmias. (Andelova et al. 2020; Danik et al. 2004)”). Also, we addressed the comment on the effect of uremic toxins on myocardial connexin43 expression and topology by another paragraph (line 291-294: “Furthermore, indoxyl sulfate decreased gap-junction intercellular communication by downregulation of Connexin 43 in cultured neonatal rat cardiomyocytes leading to increased spontaneous contractions (Changchien et al., 2020).“

  • It should be taken into consideration that ion channels disturbances trigger transient arrhythmias, while disorders of connexin channels underlie arrhythmogenic substrate for malignant re-entry arrhythmias (see f.e. Valderrábano 2007, Hoagland 2019, Saffitz 2000, Danik 2004, Salameh 2005, Dhein 2014, Thomas 2018, Dhein 2021), 

Authors: Thank you for this comment. As mentioned above, we have addressed the point with an additional paragraph (line 74-77: “Functional limitations of the conduction velocity or conduction block can result, for example, from a decrease in the conductivity of the cell-cell contacts (e.g. Connexin 43) at the intercalated disks or from prolonged recovery time facilitating life-threatening re-entry arrhythmias (Andelova et al. 2020; Danik et al. 2004).”).

  • Besides, connexin43 hemichannels activation along with pro-inflammatory signalling may be proarrhythmic (Andelova 2021). In this context to examine efficacy of treatment with SGLT2 inhibitors would be very interesting as also noted in Zoccali et al 2023 and Andelova et al. 2022).”

Authors: We thank the reviewer for the important suggestion that SGLT2 inhibitors might be a possible treatment of arrhythmias in CKD patients, so we have added a phrase. (line 340-344: “In another mouse model with prolonged QT time by induction of metabolic syndrome, Jhuo et al. found that connexin 43 expression was reduced in the ventricle. SGLT2i treatment increased the expression of connexin43, which was accompanied by a significant shortening of the QT interval highlighting SGLT2i as treatment option for malignant re-entry arrhythmias in CKD (Jhuo et al., 2021; Andelova et al. 2020).”)

  • Title Consider, please, to modify by including arrhythmia risk. 

Authors: We added "arrhythmia-promoting" to the title so that the full title is now "Impact of impaired kidney function on arrhythmia-promoting cardiac ion channel regulation" (line 1-2).

  • Abstract exhibits several typing errors using ;  that should be deleted.

Authors: Thank you, we corrected the typing errors (line 11, 14, 15, 16).

  • Key words: should be more precisely selected and should include connexin43

Authors: Thank you for the comment. The keywords "acidosis" and “autonomic nervous system” as well as “connexin43” have been added (line 22).

  • In subheading 1.1. The text in all paragraphs should be cited with most appropriate reference. Moreover, electrical instability is promoted not only by ion channels disorders but also by disorders in myocardial conduction. The latter is strongly affected by intercellular connexin channels at the gap junction located at the intercalated disc of the cardiomyocytes (see f.e. Valderrábano 2007, Kotadia 2020)

Authors: We supplemented appropriate references for the paragraphs (line 60, 67, 70, 77, 92, 184, 189, 246). As stated above, we have added another paragraph to the valuable comment that connexin channels can also cause a change in myocardial conductance (line 74-77: “Functional limitations of the conduction velocity or conduction block can result, for example, from a decrease in the conductivity of the cell-cell contacts (e.g. Connexin 43) at the intercalated disks or from prolonged recovery time facilitating life-threatening re-entry arrhythmias.” (Andelova et al. 2020; Danik et al. 2004).”)

  • Subheading 3.1: Downregulation and/or inhibition of Cx43 channels contribute significantly to electrical uncoupling of the cardiomyocytes promoting slowing and-or block of conduction facilitating life-threatening re-entry arrhythmias (see f.e. Danik 2004 as well as others). Furthermore, acidosis and Ca2+ overload also inhibit myocardial connexin43 channels and contribute to electrical uncoupling of cardiomyocytes resulting in slowing of conduction that facilitates re-entrant arrhythmias (see f.e. Salameh 2005)

Authors: Thank you very much for the excellent addition. We added another paragraph to address connexin 43 regulation in the setting of acidosis (line 167-169: “In addition, acidosis can worsen myocardial conductance by inhibiting the conductance of gap-junctions such as connexin-43 (Ek-Vitorin et al, 1996), which promotes the development of malignant re-entry arrhythmias (Rodríguez et al, 2021; Salameh et al, 2005).”)

  • Subheading 3.2.: First paragraph should be more rigorously cited. Moreover, inflammation and oxidative stress downregulate conexin43 protein (see f.e. Andelova 2022 as well as others) and deteriorate the function of connexin43 channels that contribute to electrical instability and increases susceptibility of the heart to malignant arrhythmias. Moreover, beta-adrenergic stimulation down-regulate connexin43 and alters its cardiomyocyte topology (shown by Szeiffova Bacova 2020 a,b) that is pro-arrhythmic and increase susceptibility of the heart to malignant arrhythmias (Szeiffova Bacova 2020a,b).

Authors: Thank you for this addition. We added citations in line 60, 67, 70, 77, 92 and another paragraph regarding beta-adrenergic stimulation on connexin 43 regulation (line 231-233: “Besides triggered activity, beta-stimulation in rats showed that connexin 43 was redistributed (Szeiffova et al, 2020), which is known to promote electrical instability and life-threatening arrhythmias (Peters et al, 1996).”

  • Conclusion should also involve possible implication of connexin43 channels in development of life-threatening re-entrant arrhythmias in CKD.

Authors: Thank you for this comment. We added in our conclusion the following (line 371-375: “These adaptations to CKD, in addition to predisposing to reentry tachycardia by conduction velocity alterations for instance through altered connexin 43 expression, lead to EADs via action potential prolongation and to DADs via spontaneous calcium release via activation of the NCX. EADs and DADs increase the risk for arrhythmias.”)